# Windows to the Social Mind: What Eye-Tracking Reveals About Theory of Mind in Children and Young Adults with Autism Spectrum Disorder (ASD)

**DOI:** 10.3390/bs15121622

**Published:** 2025-11-25

**Authors:** Sobh Chahboun, Brian Sullivan, David Saldaña, Mila Vulchanova, Martina Micai

**Affiliations:** 1Department of Pedagogy, Section Special Education, Queen Maud University College, 7044 Trondheim, Norway; 2Department of Psychological Science, University of Bristol, Bristol BS8 1QU, UK; brian.sullivan@bristol.ac.uk; 3Tobii Technology AB, 182 53 Danderyd, Sweden; 4Department of Developmental and Educational Psychology, Universidad de Sevilla, 41004 Seville, Spain; dsaldana@us.es (D.S.); martina.micai@iss.it (M.M.); 5Language Acquisition and Language Processing Lab, Norwegian University of Science and Technology, 7034 Trondheim, Norway; mila.vulchanova@ntnu.no; 6Coordination and Promotion of Research, Istituto Superiore di Sanità, 00161 Rome, Italy

**Keywords:** Autism Spectrum Disorder, theory of mind, eye tracking, social cognition, cognitive development, social perception

## Abstract

Human social life is dependent on the ability of individuals to understand other people as separate cognitive agents, capable of thought independent from themselves. This understanding and the attribution of mental states to others, often called Theory of Mind (ToM), is a naturally developing ability. Individuals with Autism Spectrum Disorder (ASD) seem to experience difficulty in attributing mental states to others, and this may explain impaired social interaction and communication behaviors. The Frith-Happé animations are short videos designed to test ToM development by varying the degree of intentionality present and asking viewers to describe their interpretation. The present study recorded eye movements and verbal descriptions in 15 children and 23 young adults with ASD and 20 and 15 typically developing (TD) peers, respectively. The results showed eye movement patterns in ASD and TD children did not differ significantly, but both groups differed from adults in their verbal responses. Children with Autism Spectrum Disorder (ASD) produced shorter (F(1,60) = 5.8, *p* = 0.019) and less appropriate (F(1,60) = 4.4, *p* = 0.04) ToM descriptions than TD peers, although their eye movement patterns were comparable to those of TD children. While low-level visual processing may be intact in individuals with ASD, challenges with social cognition and verbal expression may remain.

## 1. Introduction

Autism Spectrum Disorder (ASD) is defined in the DSM-5 (Diagnostic and Statistical Manual of Mental Disorders-5; [3]) as a pervasive neurodevelopmental disorder characterized by impaired social interaction and communication, and repetitive patterns of behavior. Social competences have been shown to be correlated to theory of mind (ToM) ([26]; [4]; [15]; [21]). ToM refers to the intuitive ability of recognizing one’s own and other people’s perspectives, thoughts or mental states, and the skill of being able to predict a concrete mental status ([1]; [4]; [26]; [30]; [33]). It has been claimed that atypicality in ToM is a core feature that contributes to the social and communicative challenges faced by individuals with ASD (e.g., [4]; [15]; [14]; [21]; [26]; [18]).

Testing ToM gained interest over the last few decades, but different measures have been used, from conventional psychometric measures, such as intelligence quotient tests, to behavioral measures, such as eye tracking ([23]). A common way in which ToM has been investigated is through false belief tasks, but these have limitations in involving other faculties, such as executive function and inhibitory control, as well as heavy reliance on language ability ([12]). [1] ([1]) sought to develop a new measure that would specifically tap into ToM and avoid confounds based on [16]’s ([16]) animations which showed that even the movements of geometric shapes could be perceived as intentional action. The classic false-belief task typically involves understanding that another person can hold a belief that differs from reality—for example, when an object is moved without the protagonist’s knowledge. However, successful performance depends not only on mental-state reasoning but also on language comprehension, working memory, and inhibitory control, which can confound the measurement of Theory of Mind abilities, particularly in individuals with ASD ([10]).

Following [16] ([16]), [1] ([1]) created a task involving watching the anthropomorphic motion of geometric shapes, with the prediction that individuals with ASD would be impaired in their assessment of the videos. They created 16 videos (~30 s each) involving a red and a blue triangle moving on a screen. The design included three types of videos (four of each type) divided into random movement, goal-directed and ToM sequences. Random movements could be described by surface level characteristics, e.g., the triangles are bouncing like billiard balls, whereas ToM movies might require insights into the ‘mind of the triangles’, e.g., if one triangle is behind a door, the other triangle cannot ‘see’ it. [1] ([1]) tested children with ASD, children with moderate learning disability, typically developing (TD) children, and adults. While the children with ASD performed normally on classical false-belief tasks, they performed worse than both other child groups in the experimental ToM movies. Children with ASD tended to refer to mental states that did not fit the circumstances. One reason for this might be that the non-verbal nature of the videos deprives the group with ASD of explicit verbal information from which to understand the “mindset” of the triangles. [1] ([1]) point out that the verbal component of the test is a limitation, as it is subjective and time-consuming to code and that it would be preferable to devise a non-verbal alternative.

Since the original publication, numerous studies have used the Frith-Happé animations, but with this limitation of relying on verbal descriptions. [35] ([35]) developed a more objective approach to capture spontaneous, non-verbal indicators of social attention and mental state attribution bytracking the gaze. By revealing how individuals allocate visual attention to intentional versus random motion, eye-tracking allows researchers to infer implicit ToM processing, beyond verbal or explicit responses. Participants were shown Frith-Happé animations but were asked to give a running verbal description throughout and were afterwards asked to place the actions they observed into categories of no interaction, physical interaction, and mental interaction. The results were similar to previous studies, with the new evaluation system appearing to be as accurate as the older one and taking less time to administer. Mentalizing—the ability to understand others’ thoughts, intentions, and emotions—may rely more on general cognitive abilities related to perceiving and interpreting movement rather than on traits specific to autism ([36]). This suggests that differences in mentalizing skills could be linked to broader perceptual and cognitive processes, rather than being solely determined by an autism diagnosis.

Eye tracking can also serve as a more objective measure of participant behavior while watching the Frith-Happé animations. [19] ([19]) tracked the eye movements of TD adults and examined three measures across the movie types: the average fixation durations on either triangle, the proportion of time spent on triangles versus other objects, and the ratio between the proportion of time looking at the blue triangle versus the red. They found that ToM animations elicited longer fixation times than goal-directed scenes, and goal-directed more than random ones. [39] ([39]) tracked the eye movements of 12 TD adult participants watching the Frith-Happé animations and replicated Klein et al.’s results. Furthermore, they introduced two more analyses including the proportion of time spent looking at either triangle or the ratio of time spent fixating the blue triangle versus the red. They found that the total time spent looking at either triangle increased from random to goal-directed to ToM videos, and that the proportion of time fixating the blue vs. red triangle was biased toward blue but shifted toward 50/50 as intentionality increased. These results demonstrated that mentalizing appears to be reflected in several types of gaze patterns, in addition to the verbal descriptions previously studied.

[41] ([41]) eye-tracked adults watching the animations including participants with ASD (*n* = 19) and a control TD group (*n* = 18). They investigated whether individuals with ASD would show the same fixation durations and locations as the control group. Across groups, fixation duration and fixation proportion measures were similar, but the participants with ASD used inappropriate descriptions of what they had seen. They hypothesized that individuals with ASD have the basic building blocks of social gaze interaction intact, and that social impairments are located in higher functions beyond gaze control. In a second experiment, they investigated whether the participants could take the hypothetical “perspective” of the triangles into account. A dot would appear, and the participants were asked to indicate whether this appeared on the triangle’s left or right. This would sometimes be congruent, and sometimes incongruent, with the perspective of the participant. There was no significant difference between the groups on this, suggesting that taking another’s visuo-spatial perspective is separate from mentalizing. It is unclear why this is the case, but one explanation may be that, in evolutionary terms, it was necessary to know whether one was within a predator’s visual field long before it was necessary to be able to infer mental states.

Additionally, while the studies described above have shown that fixation durations and proportions correlate with the intentionality present in the animations, these results may alternatively indicate responses to low-level motion cues that do not actually reflect ToM development. [31] ([31]) explored the kinematic characteristics of the Frith-Happé animations by tracking the amount of motion made by each triangle in the set of animations. They found that, in the higher intentionality videos, the triangles moved less and were more often close to each other. Because duration of motion and distance between triangles were not controlled for across videos, they devised two new metrics that would try to account for these discrepancies, including mobile triangle time, the proportion of time spent looking at either triangle when they were mobile, as this controls for the fact that mobile time varies across movie types (using distance between triangles as a covariate). Additionally, blue and red triangle mean pursuit durations were calculated by averaging the time spent pursuing each triangle. Note, they define pursuit as any unbroken period looking at a triangle, even including moments from mobile to non-mobile triangles. In these cases, the period of non-mobility is excluded, and the pursuit portions are summed and used in the average across all pursuit periods.

[31] ([31]) replicated prior findings that triangle fixation durations and the proportion of time spent on the triangles increased with intentionality. Furthermore, they showed that mobile triangle time increased, and that blue and red triangle pursuit durations also increased with intentionality. From these results they conclude that, while the prior metrics used by other researchers are confounded by motion cues versus intentionality, their new metrics do provide evidence that there are gaze differences as intentionality increases in the animations, even when controlling for motion cues. They note that this may potentially explain [41]’s ([41]) lack of a significant difference between ASD and TD groups.

Recent work has moved beyond passive viewing paradigms toward more ecologically valid, interactive approaches. For instance, [38] ([38]) used interpersonal eye-tracking during live social exchanges to show how gaze coordination, pupil dilation, and blinking dynamically align between partners, revealing moment-to-moment coupling of attention and mental-state understanding in real interaction contexts. Additionally, [13] ([13]) used eye-tracking in a visual perspective-taking task to show that greater spatial disparity increased gaze shifts between agents and objects. Their results also demonstrated that social-context cues, such as the agent’s characteristics, modulate attention during perspective-taking.

For the current study, we had three primary goals. First, we aimed to replicate the validity of the measures introduced by [19] ([19]), [39] ([39]), and [31] ([31]) that control for stimulus motion. Second, we sought to determine whether these measures reveal differences between young adults and children, particularly in the motion-controlled metrics, given that ToM is still developing in children. Third, we examined whether ToM scores derived from verbal responses would be the strongest predictor of eye movement strategies above other variables such as age, diagnostic group, or motion-controlled eye-tracking metrics, with individuals with ASD expected to show lower scores in this regard. Alternatively, if no group differences were found, this would suggest that the previously reported ToM differences between individuals with ASD and the TD control group for verbal descriptions of the animations may reflect independent processes not related to eye movement strategies.

Moreover, [11] ([11]) explain that the concept of Autism Spectrum Disorder (ASD) has evolved internationally—highlighting the heterogeneity of presentation, diagnostic challenges, and issues of inclusion. Their work underscores the need to consider broader cognitive and linguistic differences alongside social-communication symptoms when studying ASD, thereby reinforcing the value of non-verbal, attention-based methods such as eye-tracking in research on social cognition.

## 2. Materials and Methods

### 2.1. Participants

Participants with ASD and TD, all Spanish native speakers, were recruited. The first age group included children in the age range of ten to thirteen years (ASD: *n* = 15, $\bar{x}$ = 11.17; TD: *n* = 20, $\bar{x}$ = 12.26). The second age group included adolescents and young adults from fourteen to twenty-three years old (ASD: *n* = 23, $\bar{x}$ = 20.44, TD: *n* = 15, $\bar{x}$ = 20.6). We selected these two concrete age ranges because, in typical development, the acquisition of figurative language reaches a peak around ten to twelve years old, and this knowledge is usually consolidated around late teenage/young adulthood. Previous research has shown that the comprehension of pragmatic language develops gradually through childhood and adolescence, with a marked increase around the preteen years, and continues to refine into young adulthood ([28]; [34]; [6]).

Participants or their legal guardians provided signed consent for entry into the study. Children with ASD had all previously been diagnosed according to criteria in the Diagnostic and statistical manual of mental disorders ([2]) by a psychologist or psychiatrist with expertise in autism. The diagnosis of ASD was confirmed according to the Autism Diagnostic Observation Schedule (ADOS, [27]) by a trained psychologist.

None of the participants had diagnosed comorbid conditions such as language delay, or other neurodevelopmental disorders, based on parent report and clinical documentation. All participants demonstrated cognitive and verbal abilities within the average range for their age group, as verified through school records and prior standardized assessments. Thus, the potential confounding effects of general cognitive ability or language impairment on ToM performance were minimized.

### 2.2. Animation Task

The present study used the 12 animation sequences plus 3 training animations, from [1] ([1]). The animations included three distinct types of motion: random movement, goal-directed interactions, and ToM interactions. In total, there were four movies for each type in the test phase, and one of each, during the training phase. Two triangles, one big and red and the other blue and small, appeared in the videos moving in a variety of ways, e.g., from bouncing like billiard balls to apparently chasing one another.

A Tobii T120 (Tobii AB, Danderyd, Sweden) was used to display the animations and record eye movements. MATLAB (R2014b) (Matrix laboratory, Mathworks, Natick, MA, USA) and the Psychophysics Toolbox ([7]; [29]; [20]) were used to create the experimental program that displayed the animations to the participants while their eye movements were recorded and to record audio of their verbal descriptions after each animation for later coding and analysis.

### 2.3. Procedure

The eye tracking session was performed in a single session in a quiet, distraction-free room. Participants were seated in front of the eye tracker at about 60 cm from the screen displaying the videos, the screen subtended ~32° × 26° and refreshed at 60 Hz. The videos were displayed at 640 × 480 and subtended ~16° × 12°. As part of the tracker calibration, seating was adjusted as needed to ensure participants were centered in the eye tracker’s field of view. The tracker was calibrated by having the participant look at a calibration point (a white circle ~1° in diameter with a central black circle ~0.2° diameter).

To ensure participants were familiar with the task, three practice animations—one from each movement category—were shown at the beginning of the session. Participants watched each animation and were asked to verbally describe what they believed the triangles were doing. Each animation was presented only once per participant, with the experimenter verbally emphasizing the importance of close attention. During the testing phase, the animations were presented in a randomized order across participants.

The primary eye-tracking variables analyzed included mean fixation duration, saccade length, dwell time on predefined areas of interest (AOIs, e.g., agents, objects, background), and gaze transitions between agents and targets. Each animation lasted approximately 40 s and was presented at a frame rate of 30 frames per second. The order of animation presentation was counterbalanced across participants to control for potential order or fatigue effects. These variables were chosen to capture both attention allocation and dynamic gaze coordination relevant to ToM processing.

### 2.4. Verbal Description Scoring

Participants’ verbal descriptions of each animation were manually scored by two psychologists blind to the group (SC, MM) according to the following three criteria: intentionality, relevance, and length of the answer ([9]).

Intentionality: This score reflects the use of intentionality in mentalist terms. The degree of attribution of mental states to the triangles (agents) in the animations was estimated by analyzing the description provided by the participants. To control for subjectivity in the assessment of language, the analysis was based on the use of the verbs in each phrase to describe the actions of the triangles. The degree of intentionality reflected in every action mentioned was measured on a numeric scale ranging from zero to five, created specifically for this task. The scale rates language as progressing from low to high intentionality, ranging from the recognition of movement of an agent, to describing the actions of both agents, to describing their mental states.

The scale-score ([9]) captured a progression from basic motion perception to increasingly complex interpretations of agency, interaction, and mental state attribution. The structure is rooted in established frameworks of social cognition and mentalizing, which propose that understanding others’ actions progresses from simple movement perception to intentionality and, ultimately, to mental state inference.

The distinctions between levels reflect this theoretical gradient: the lower levels (0–2) focus on physical movement and goal-directed action, while the higher levels (3–5) incorporate social interaction and mental state attribution.

Score 0: Participants did not perceive agency, actions, or mental states. The agent’s movement appeared random and lacked intention or interaction, such as “moving from one side to another” or “floating.”

Score 1: The agent acted with a purpose or goal but without interacting with another agent; for example, “walking” or “swimming”.

Score 2: The agent acted with a purpose in relation to another agent, but their actions were parallel in time. Examples include “fighting” or “following.”

Score 3: The agent interacted with another agent in response to their actions, making the interaction sequential in time. Examples include “chasing” or “holding.”

Score 4: The agent’s actions reflected an understanding of another’s mental state, as seen in behaviors like “discussing,” “meaning,” or “encouraging.”

Score 5: The agent actively influenced or manipulated another’s mental state, as in “simulating,” “cheating,” or “convincing.”

Appropriateness: Appropriateness measured the understanding of the events described in the animations, according to the intentions of the designers. The score, ranging from zero to two, was based on the script prepared for each animation based on prior studies ([9], [8]; [32]). The degree of relevance was calculated by analyzing the actions and interactions of different agents. For example, in one animation, the large red triangle “convinced” the small blue one to leave. An appropriate description needed to convey that the small blue triangle was reluctant to go out and that the large red triangle attempted to “persuade” or “convince”. A less appropriate description focused on an aspect of history or only one of the characters; for example, noting that only the small blue did not want to leave, or that the large red triangle pushed the small blue triangle out. Participants may have also made improper disclosures, i.e., statements that did not relate to important events and instead consisted of comments on an aspect of the sequence; for example, “I do not like the triangles”. A score of zero occurs in the case that no description occurs, or no mention of any onscreen events was made.

Length of verbal answer: The length of verbal answers was scored based on the number of sentences the participant used to describe the animation: 4 = more than four sentences, 3 = four sentences, 2 = two sentences, 1 = one sentence and 0 = no response.

## 3. Results

### 3.1. Verbal Scores Statistical Analysis

Repeated measures 2 × 3 ANOVA analysis (Movie Type × Group) was conducted on subjects’ verbal scores, using Bonferroni correction applied for multiple comparisons. When the omnibus test revealed a significant main effect or interaction (α < 0.05), we conducted follow-up post hoc comparisons—either one-way repeated measures ANOVA or paired *t*-tests—to explore the source of the effect. Verbal scores for each animation were averaged across the two independent raters, and animations were grouped by category (Goal-Directed, Random, ToM). The mean score for each category was calculated for each participant. This yielded three score types (Intentionality, Length, and Appropriateness) for each of the three movie types, which were analyzed separately.

Table 1 shows the means for all verbal score types.

#### 3.1.1. Intentionality

Overall analysis of intentionality scores revealed a main effect of movie type, F(2,59) = 156, *p* < 0.001, with means increasing from random (0.74) to goal-directed (2.30) to ToM (3.30). Post hoc paired *t*-tests (Bonferroni-corrected) confirmed that all three types differed significantly: ToM > goal-directed, t(63) = 15.8, *p* < 0.001; ToM > random, t(63) = 18.4, *p* < 0.001; and goal-directed > random, t(63) = 11.9, *p* < 0.001.

No main effect of group (ASD vs. TD), F(1,60) = 0.43, *p* = 0.51, or age, F(1,60) = 0.07, *p* = 0.79, was found. There was a significant movie type × group interaction, F(2,59) = 3.37, *p* = 0.041, and a movie type × age interaction, F(2,59) = 3.59, *p* = 0.034; the three-way interaction was not significant, F(2,59) = 1.07, *p* = 0.35.

Post hoc comparisons for the movie type × group interaction indicated that in the random condition, participants with ASD scored higher in intentionality than TD participants (0.98 vs. 0.50), F(1,60) = 5.4, *p* = 0.02.

For the movie type × age interaction, ToM animations elicited higher intentionality scores in children than in young adults (3.56 vs. 3.13), F(1,60) = 5.13, *p* = 0.027.

Further exploration of the three-way pattern showed that in random movies, TD children scored higher than children with ASD (1.16 vs. 0.65), F(1,60) = 4.31, *p* = 0.042. In ToM movies, TD children also tended to score higher than children with ASD (3.40 vs. 2.90), F(1,60) = 3.92, *p* = 0.052, and scored higher than young adults with ASD (3.60 vs. 2.90), F(1,60) = 5.8, *p* = 0.019.

To summarize, post hoc comparisons examining the movie type × group interaction showed that participants with ASD had higher intentionality scores than TD participants for random movies (0.98 vs. 0.50), F(1,60) = 5.4, *p* = 0.02.

Post hoc comparisons for the movie type × age interaction found that young adults used less intentionality language on ToM movies compared to children (3.13 vs. 3.56), F(1,60) = 5.13, *p* = 0.027.

Intentionality language varied strongly by movie type, being highest for ToM movies. TD children generally produced more intentionality-based descriptions than children with ASD in ToM animations, while ASD participants scored higher in the random condition.

#### 3.1.2. Appropriateness

Overall analysis on appropriateness revealed a main effect of movie type F(2,59) = 19.2, *p* < 0.001. On average, participants had appropriateness scores of 1.4, 1.6, and 1.1 for random, goal-directed and ToM movies, respectively. ToM videos’ appropriateness score was also significantly different compared to goal-directed (t(63) = 2.7, *p* = 0.024) and random (t(63) = 6.8, *p* = 0.001) ones. No statistical difference was detected between random and goal-directed scores (t(63) = 2, *p* = 0.14).

No significant interactions between movie type and group (ASD vs. TD), F(2,59) = 1, *p* = 0.9, between movie type and age, F(2,59) = 1.9, *p* = 0.17, and between all three movie types, F(2,59) = 0.74, *p* = 0.48, were observed.

Post hoc comparisons (exploratory) indicated:In random movies, children with ASD tended to score higher than TD children (1.5 vs. 1.1), F(1,60) = 3.3, *p* = 0.074.In ToM movies, children with ASD scored lower than TD children (0.83 vs. 1.14), F(1,60) = 4.4, *p* = 0.04.In ToM movies, children with ASD also scored lower than young adults with ASD (0.83 vs. 1.2), F(1,60) = 7.1, *p* = 0.01.

These age-related differences in ASD groups are consistent across conditions but do not reflect an interaction with movie type.

Appropriateness scores were generally lowest for ToM animations. TD children outperformed children with ASD in ToM movies, while age-related differences in ASD groups suggested higher scores in young adults compared to children.

#### 3.1.3. Length of Verbal Answer

Overall analysis revealed a main effect of movie type, F(2,59) = 39, *p* < 0.001, with mean scores of 2.4 (random), 2.65 (goal-directed), and 3.15 (ToM). All three types differed significantly: R < GD, t(63) = 3.6, *p* = 0.003; R < ToM, t(63) = 8.4, *p* < 0.001; GD < ToM, t(63) = 7.5, *p* < 0.001.

There was also a main effect of age within the ASD group, F(1,60) = 6.6, *p* = 0.013, with children with ASD producing shorter answers overall compared to young adults with ASD (2.39 vs. 2.95).

A movie type × group interaction was observed, F(2,59) = 4.3, *p* = 0.018; however, follow-up simple effects showed no consistent group differences within a specific movie type after correction for multiple comparisons.

Post hoc comparisons showed:In ToM animations, children with ASD had shorter responses than TD children, F(1,60) = 5.8, *p* = 0.019.In random animations, young adults with ASD scored marginally higher than TD young adults (2.7 vs. 2.1), F(1,60) = 3.1, *p* = 0.084.Across all movie types, children with ASD scored lower than young adults with ASD—random: 2.17 vs. 2.7, F(1,60) = 4.6, *p* = 0.037; goal-directed: 2.3 vs. 2.86, F(1,60) = 4.3, *p* = 0.04; ToM: 2.67 vs. 3.29, F(1,60) = 6.9, *p* = 0.01.

Length scores increased with movie intentionality. Children with ASD consistently produced shorter verbal descriptions than both TD children (in ToM) and young adults with ASD (across all movie types), indicating group differences by age, with young adults with ASD giving longer descriptions than children with ASD.

To summarize across all verbal measures, children with ASD scored lower than their TD peers on all three verbal measures—intentionality, appropriateness, and length—with the largest differences in the ToM condition. For example, in ToM animations, they used less intentionality-based language (2.90 vs. 3.36), gave less appropriate responses (0.83 vs. 1.14), and produced shorter descriptions (2.68 vs. 3.26). Differences in goal-directed and random conditions were smaller and less consistent. Among young adults, ASD and TD groups performed similarly across measures, with only minor differences remaining. This pattern suggests age-related improvement in verbal performance in ASD, particularly for intentionality in ToM contexts (see Table 1).

### 3.2. Eye Gaze Analysis

Prior to analysis, raw gaze data were visually inspected for signal quality. Missing samples caused by blinks or temporary loss of tracking were linearly interpolated when gaps were shorter than 150 ms; longer gaps were excluded from fixation computation. Eye-position data were median-filtered with a 9-sample (~75 ms) moving window to reduce high-frequency noise. Trials with more than 30% track loss or poor calibration accuracy were excluded from further analysis, as stated previously. The dispersion-based fixation detection algorithm (spatial threshold = 1.5° diameter; minimum duration = 100 ms) was implemented using the EyeMMV toolbox, and all fixation events were re-checked for plausibility using in-house scripts. These steps ensured that artifacts, drifts, and missing-data segments did not bias fixation duration or proportion measures. Despite our best efforts to obtain high-quality tracking data, participants became fatigued during the task. As a result, we decided to analyze trials with no more than 30% track loss; trials exceeding this threshold were excluded. Additionally, subjects had to have at least one valid trial in a movie category to be admitted into further analysis. Due to these criteria, from our original 64 participants, only 26 participants (12 ASD; 14 TD) were included in the analyses, affecting the generalizability of the results. Note that not all 26 participants had an equal number of valid trials, such that the repeated measures and pairwise comparisons presented below typically contain less than 26 participants and, as a result, have varying degrees of freedom (DOF) and descriptive statistics dependent on the trials available for comparison.

If binocular eye data were present, the left and right eye signal were averaged; otherwise, only the monocular signal was used. These data were then median filtered using a window size of 9 samples (~75 ms).

Frame-by-frame eye tracker data were labeled into three categories: blue triangle, red triangle, and other. Using an in-house Matlab video annotation tool, the positions of both triangles were manually coded by recording the position of the three corners of each triangle on each frame. Due to variation in the manual annotation, the three corners were averaged together for an estimate of the center of mass of the triangle.

To provide a label, on each frame, the eye gaze position was evaluated to test if it was within 1° of a bounding box that was fitted to the most extreme points of the triangle. This is a liberal criterion and was chosen due to varying eye data quality in our subjects. If the current gaze location fell within this bound, then it was labeled as being on either the blue triangle, the red triangle, or both. In the event that the gaze was unlabeled, it was considered part of the ‘other’ category. Because the videos (~15 Hz) were much slower than the eye tracking data, multiple eye tracker coordinates were evaluated per frame of triangle coordinates.

Following the procedure of [39] ([39]), we segmented our data into fixations via the dispersion algorithm (spatial threshold of 1.5° diameter, and minimum duration of 100 ms) using the implementation in the Eye Movements Metrics & Visualizations (EMMV) Matlab Toolbox ([22]). The fixations produced from this were given object labels by evaluating the frame-by-frame labels within the duration of each fixation. This range of frame-by-frame data was examined to check that at least one frame had a label for either triangle or both, and if so, the entire fixation was labeled as blue, red, or both triangles, dependent on the labels present; otherwise, the fixation was marked as ‘other’.

#### 3.2.1. Fixation Duration

Using repeated measures ANOVA, we tested the effects of movie type (Random, Goal-Directed, and ToM), group (ASD vs. TD), and age (child vs. young adult) on predicting fixation duration. There was only a significant main effect of movie type (F(2,15) = 19.9, *p* < 0.001), compared with movie type × group (F(2,15) = 0.6, *p* = 0.54), movie type × age (F(34,10) = 0.42, *p* = 0.66), and movie type × Age × group (F(2 × 15) = 1.5, *p* = 0.25). Respectively, the durations per movie type were 393, 381, and 477 ms, for Random, Goal-Directed, and ToM movies. Pairwise analysis showed that while Random and Goal-Directed movies had similar fixation durations, t(19) = 0.36, *p* = 0.75, ToM was different from Random movies, t(21) = 7.3, *p* = 0.75, and Goal-Directed movies, t(20) = 4.1, *p* = 0.001 (Figure 1).

#### 3.2.2. Fixation Proportion on Any Triangle

Using the same procedure, we examined the effects of movie type on the proportion of fixations on either triangle during the trials. There was only a significant main effect of movie type (F(2,15) = 25.05, *p* < 0.001) and no interaction effects. None of the following passed the alpha 0.05 threshold: movie type × group (F(2,15) = 2.42, *p* = 0.12), movie type × age (F(34,10) = 0.61, *p* = 0.56), movie type × age × group (F(2,15) = 1.01, *p* = 0.39) (Figure 2).

Comparing across movie types, via paired *t*-tests, we observed that all movies differed from one another: Random and Goal-Directed differed marginally, 78.2% vs. 82.5%, t(19) = 1.9, *p* < 0.07; Random and ToM, 78.3% vs. 92.1%, t(21) = 7.31, *p* < 0.001; and Goal-Directed and ToM, 79.2% vs. 90.3%, t(20) = 6.6, *p* < 0.001.

#### 3.2.3. Ratio of Fixations on Blue Versus Red Triangle

Using the same procedure as above, we tested the effect of movie type (Random, Goal-Directed, and ToM), group, and age on predicting the ratio of fixations allocated to the blue or red triangle, calculated as the time of fixation on blue, divided by the sum of time fixating both blue and red.

We found a significant main effect of movie type (F(2,15) = 19.2, *p* < 0.001). There was no significant interaction of movie type × age (F(2,15) = 0.31, *p* = 0.74), and for movie type × group (F(2,15) = 0.34, *p* = 0.71). Furthermore, there was no statistically significant interaction between movie type × age × group (F(2,15) = 1.5, *p* = 0.25).

Comparing across movie types, via paired *t*-tests, we found no difference between Random and Goal-Directed movies, 49.7% vs. 48.3% t(19) = 0.68, *p* = 0. 5. However, there were differences in both Random and ToM movies, 50.8% vs. 44.17%, t(21) = 3.02, *p* = 0.006, and Goal-Directed and ToM movies, 47.75% vs. 42.75%, t(20) = 4.02, *p* = 0.001 (Figure 3).

#### 3.2.4. Mobile Triangle Time

A Generalized Linear Model (GLM) using movie type, age, and group was used to predict the proportion of time spent fixating the mobile triangle (Figure 4).

We observed a main effect of movie type on the total mobile triangle time duration, F(2,15) = 8.09, *p* < 0.004. A pairwise *t*-test showed no difference between Random versus Goal-Directed movies, t(19) = 0.49, *p* = 0.63, (79.5% vs. 78.7%). However, pairwise *t*-tests did show differences between Random and ToM movies, t(21) = 3.43, *p* = 0.003, (79.7% vs. 85%), and Goal-Directed and ToM movies, t(20) = 4.01, *p* = 0.001, (76.5% vs. 83.2%).

There were no further interaction effects of movie type with group (F(2,15) = 1.92, *p* = 0.18), age group (F(2,15) = 0.16, *p* < 0.86), or both age group and movie type (F(2,15) = 0.58, *p* = 0.58).

#### 3.2.5. Mean Smooth Pursuit Duration

A GLM using movie type, triangle type, age group, and group was used to predict average smooth pursuit durations. For average pursuit duration, we observed a main effect of movie type (F(2,21) = 21.35, *p* < 0.001), and a significant effect of triangle type F(2,21) = 3.06, *p* = 0.001) with a trend for the red triangle to be pursued longer (~200 ms) (Figure 5).

Table 2 summarizes the results of the interactions. Only a significant interaction between movie type and triangle type was found (F(2,15) = 6.25, *p* = 0.011). Pairwise *t*-tests showed a small effect for participants to pursue the blue triangle longer in ToM movies, 1.35 s v. 1.8 s, (t(22) = 4.45, *p* < 0.001). There was no effect for Random movies, (t(21) = 1.1, *p* = 0.29) nor Goal-Directed movies (t(20) = 1.77, *p* = 0.1).

### 3.3. Regression Analysis for Eye Gaze and Verbal Responses

We analyzed the relationship between verbal scores and eye movements by looking at the main effects in a regression analysis using intentionality, appropriateness, and length of verbal responses as predictors for each type of eye movement behavior for the same participants with both verbal responses and eye gaze data. If the full model was considered significant, we examined the components further via simple means testing on the regression coefficients.

#### 3.3.1. Average Fixation Duration

We observed a significant relationship between the average fixation duration for random animations and the appropriateness, intentionality, and length of verbal responses, F(3,19) = 5.81, *p* = 0.005. This effect was mediated entirely by length, t(21) = 3.7, *p* = 0.002, n^2^ = 0.38, β = −0.65. There was no significant relationship between look proportion and verbal response measures (appropriateness, intentionality, or length) for goal-directed animations, F(3,18) = 0.87, *p* = 0.47, or for ToM animations, F(3,20) = 0.72, *p* = 0.55.

#### 3.3.2. Proportion of Looks to Any Triangle

No statistically significant relationship was found between look proportion and appropriateness, intentionality, or length of verbal responses for random animations, F(3,19) = 0.39, *p* = 0.76, goal-directed animations, F(3,18) = 0.55, *p* = 0.66, or ToM animations F(3,20) = 0.28, *p* = 0.84.

#### 3.3.3. Blue and Red Triangle Fixation Proportion Ratio

No statistically significant relationship was found between blue and red triangle fixation proportion ratio for appropriateness, intentionality, or length of verbal responses for random animations, F(3,19) = 0.33, *p* = 0.8, goal-directed animations, F(3,18) = 0.41, *p* = 0.75, or ToM animations, F(3,20) = 0.27, *p* = 0.84.

#### 3.3.4. Mobile Triangle Time

No statistically significant relationship was found between mobile triangle time for appropriateness, intentionality, or length of verbal responses for random animations, F(3,19) = 0.39, *p* = 0.76, goal-directed animations, F(3,18) = 0.65, *p* = 0.59, or ToM animations, F(3,20) = 0.65, *p* = 0.6.

#### 3.3.5. Mobile Pursuit Duration

No statistically significant relationship was found between mean pursuit duration on the blue triangle for appropriateness, intentionality, or length of verbal responses for random animations, F(3,19) = 0.36, *p* = 0.78, goal-directed animations, F(3,18) = 0.36, *p* = 0.78, or ToM animations, F(3,20) = 1.06, *p* = 0.39.

A marginally statistically significant relationship was found between mean pursuit duration on the red triangle for random movies, F(19,3) = 2.79, *p* = 0.07. This was solely due to measures of the length of verbal responses, whereby increased pursuit of the red triangle resulted in shorter descriptions, *B* = −0.13, t(21) = 1.98, *p* = 0.06.

No relationship between mean pursuit duration on the red triangle was found for appropriateness, intentionality, or length of verbal responses for goal-directed animations, F(3,18) = 0.5, *p* = 0.69, or ToM animations, F(3,20) = 0.67, *p* = 0.58.

## 4. Discussion

The present study aimed to gather eye-tracking data while participants watched the Frith-Happé animations, including both children and young adults with ASD and their TD peers. The data were analyzed using the original measures introduced by [19] ([19]), [39] ([39]), and [31] ([31]), which account for stimulus motion. In addition, ToM verbal scoring methods were employed to assess ToM development in a cross-sectional design.

Regarding the verbal responses, as expected, we found clear effects of movie type on intentionality, appropriateness, and length scores. ToM movies received the highest scores for intentionality and length, followed by goal-directed and random movies, whereas appropriateness scores were lowest for ToM and higher for goal-directed and random movies. Across all verbal measures, younger participants with ASD tended to use less intentionality-based language, provide less appropriate responses, and give shorter descriptions compared to TD children. However, among young adults, these differences diminished, with both groups performing similarly. Similar patterns in young adults with ASD have also been observed in [37] ([37]). Differences in ToM performance among children with ASD compared to TD, but not young adults may be attributed to the accumulation of social experience over time ([17]; [37]). Furthermore, improved performance in adolescence and young adulthood may result from greater exposure to language, and improved competence in grammar, vocabulary, pragmatics, and figurative language ([37]). Furthermore, young adults with ASD may have had more exposure than children to behavioral interventions aimed at improving emotion perception, social understanding, and empathy. As [37] ([37]) suggested, future studies should incorporate age-appropriate tasks that account for these cognitive, experiential, and linguistic developments into adolescence and young adulthood.

Regarding the eye movement data, we found a main effect in the ToM movies for most of the eye movement measures (e.g., fixation duration, fixation proportion on any triangle, ratio of fixations on blue versus red triangle, and total mobile triangle time duration). This pattern confirms the validity of the experimental task to discriminate between different movie types and reflects the fact that participants needed longer visual exploration to understand the ToM scenes. ToM movies may engage higher-order cognitive processes, requiring more sustained attention to interpret the mental states of the agents involved. This suggests that participants allocate more attention to elements in the ToM movies that necessitate the identification of mental states and interactions between agents, as opposed to the simpler goal-directed or random behaviors in the other movie types. The participants’ increased attention to certain areas of interest in the ToM movies suggests that these movies elicited greater visual focus; however, the present data do not allow us to determine whether this reflects greater engagement, increased cognitive demands, or both. The dynamic nature of the ToM movies likely required participants to pay closer attention to track the movements of agents, as they interacted with one another, further emphasizing the increased complexity and sustained cognitive effort required to process these animations. The Frith-Happé animations used in this study are intentionally minimalistic, featuring line drawings with few elements and relying on motion to capture visual attention. While this simplicity allows for clear tracking of gaze behavior, it may also limit the extent to which group differences can emerge, as the cognitive demands are relatively low. In more complex scenes—those involving richer social cues, background elements, or competing stimuli—differences between groups might become more pronounced due to the increased need for interpretation and prioritization of social information. Moreover, beyond basic fixation metrics, additional measures, such as gaze-switching behavior (e.g., the frequency of shifting attention between the red and blue characters) may provide more nuanced insights into attentional strategies and social processing differences. These alternative metrics could be particularly informative in detecting subtle group-level variations that are not apparent through overall fixation duration alone.

Regression analyses examining the relationship between verbal scores and eye movements showed that, for Random movies, longer verbal descriptions were associated with specific gaze patterns. No significant relationships emerged for Goal-Directed or ToM movies. Given that the gaze analyses were based on a smaller subset of participants with complete data, these results should be interpreted cautiously, as the limited sample size may have reduced the power to detect effects in the other conditions.

Interestingly, no differences were observed in either eye movement patterns or verbal responses among young adults. This finding indicates that eye-tracking and verbal response measures may be more sensitive to developmental differences, particularly in childhood, whereas young adults may have developed more comparable cognitive and verbal abilities related to theory of mind (ToM). It is possible that group differences in eye movement patterns are more apparent in younger children under the age of 10, reflecting developmental variations in social attention or cognitive processing. By around age 10; however, these differences may diminish, with eye movement patterns between children and young adults becoming more similar. Future studies might benefit from stratifying age groups more finely to capture these potential age-related effects.

These results highlight that, while eye movement data provide valuable insights into cognitive processing, they may not fully capture the complexities of ToM, particularly in relation to verbal tasks. The observed discrepancy between eye movements and verbal responses underscores the importance of utilizing multiple assessment measures, such as eye tracking and verbal output, when evaluating ToM abilities in both children and young adults. Additionally, the findings suggest that while visual processing mechanisms may be preserved, challenges in verbal expression related to social cognition may persist, particularly in younger individuals.

The present findings can also be interpreted in terms of differences between implicit and explicit components of ToM. Classic accounts (e.g., [15]; [5]) have emphasized that ToM involves both the spontaneous, intuitive tracking of others’ mental states and the more deliberate, reflective reasoning about beliefs and intentions. The pattern observed here—comparable gaze behavior but reduced verbal ToM performance in individuals with ASD—suggests that implicit, nonverbal processes supporting social attention may be relatively preserved, whereas explicit, language-based reasoning about mental states remains more demanding. This interpretation aligns with prior evidence indicating that individuals with ASD can demonstrate sensitivity to social information at an implicit level but experience difficulties in explicitly articulating or reasoning about such information ([10]; [12]). These results therefore extend previous theoretical work by illustrating how distinct levels of ToM processing may differentially manifest across eye-tracking and verbal performance measures.

Future research should examine whether these patterns are consistent in larger longitudinal samples and across different experimental conditions. Further investigation into the cognitive mechanisms underlying gaze behavior will contribute to a more comprehensive understanding of ToM development and the factors that shape social cognition across different developmental stages.

## 5. Conclusions

This study investigated the relationship between visual processing and ToM abilities in children and young adults with ASD and their TD peers using the Frith-Happé animations. By analyzing both eye movement patterns and verbal responses, we aimed to explore how individuals process social information and whether differences in social cognition are primarily due to challenges in perceiving social cues or in articulating mental state attributions.

Our findings highlight several important insights into the development of ToM. Firstly, verbal response analyses revealed that children with ASD used less intentionality-based language, provided shorter descriptions, and demonstrated lower appropriateness scores compared to TD children. This suggests that they may experience difficulty in spontaneously verbalizing mental states and social interactions. However, these differences diminished in young adulthood, with ASD and TD adults performing similarly. This pattern suggests that exposure to social experiences, cognitive maturation, and language development over time may contribute to improvements in ToM-related verbal abilities. It also emphasizes the role of continued social and linguistic engagement in supporting ToM development. Second, eye movement data revealed a main effect of movie type, with ToM animations requiring more sustained visual attention and eliciting longer fixation durations compared to goal-directed and random animations. This finding supports the validity of the experimental task in distinguishing different levels of intentionality and highlights the increased cognitive demands associated with ToM processing. However, crucially, there were no significant differences in eye movement patterns between the ASD and TD groups. Both groups employed similar gaze strategies when observing the animations, suggesting that individuals with ASD are able to visually process social interactions comparably to their TD peers. This finding is particularly noteworthy, as it suggests that difficulties in social cognition may not stem from fundamental differences in visual attention or perception, but rather from challenges in translating these perceptions into explicit verbal descriptions.

The lack of differences in gaze behavior between groups challenges traditional assumptions that ToM deficits in ASD necessarily involve atypical visual processing of social stimuli. Instead, our findings suggest that individuals with ASD may perceive social information in a manner similar to TD individuals but may struggle with exploiting this information and verbally articulating mental states and social dynamics. This discrepancy between visual processing and verbal expression highlights the need for a multidimensional approach when assessing social cognition, one that takes into account both verbal and non-verbal indicators of ToM abilities. These results are also consistent with current findings in other domains of visual processing in autism reflecting an in-the-moment misalignment between language and visuo-spatial systems ([24], [25]).

The current results have implications for future research and clinical practice. Firstly, the findings suggest that interventions should focus not only on improving ToM understanding, but also on enhancing verbal expression and the ability to articulate social interactions. Given that verbal difficulties appear to be more pronounced in childhood, but improve with age, early interventions that support language development and social reasoning may be particularly beneficial. Secondly, they emphasize the need for assessment tools that do not rely solely on verbal descriptions, as these may underestimate the social cognitive abilities of individuals with ASD. Eye tracking and other implicit measures of social attention could provide a more comprehensive picture of ToM abilities and help refine intervention strategies aimed at improving social communication. In particular, eye-tracking can offer valuable insights into how children attend to and process social cues, even in the absence of verbal responses. This makes it a promising tool for assessing ToM in minimally verbal children, where traditional language-based assessments may be inadequate or biased. From a clinical perspective, eye-tracking can help identify atypical social attention patterns and guide individualized interventions. In educational settings, it may inform tailored support by revealing each child’s social-cognitive strengths and needs.

Future research should examine how ToM abilities emerge over time in individuals with ASD and how different factors—such as social experience, language proficiency, and cognitive training—contribute to these changes. Additionally, studies using alternative non-verbal ToM measures could help further disentangle the relationship between social perception and verbal articulation in ASD.

In conclusion, this study reinforces the idea that while individuals with ASD may experience challenges in verbally expressing mental states, their ability to visually process social interactions remains largely intact. Understanding the nuanced relationship between visuo-spatial perception, cognition, and language in social processing can help bridge the gap between observed difficulties and actual abilities, ultimately supporting individuals with ASD in navigating social interactions more effectively.

### Limitations

The present study provides valuable insights into social comprehension and processing in children and young adults; however, it has several limitations that should be addressed in future research. Firstly, the age range in the young adult group was broader compared to the children’s group. While this allowed for the inclusion of a diverse sample, it may have introduced variability in the developmental trajectories that could influence the findings. Future studies should aim to narrow the age range within each group to better identify when developmental shifts occur between childhood, adolescence, and young adulthood. A more refined age categorization could help clarify whether the observed improvements in verbal responses among young adults are driven by social experience, language proficiency, or other cognitive factors.

Secondly, limiting the analysis to trials with no more than 30% track loss resulted in a significant reduction in usable eye-tracking data. While this criterion was necessary to ensure data quality, it may have led to the exclusion of valuable information and potentially introduced selection bias. Participants who experienced greater difficulties maintaining fixation or engaging with the task may have been underrepresented in the final analysis. This data loss may have also obscured potential main effects related to age or group differences, limiting the ability to draw definitive conclusions about the relationship between eye movements and ToM abilities. Future research should explore methods to enhance data retention and improve the reliability of eye-tracking measures, such as optimizing calibration procedures, using more engaging stimuli, or incorporating alternative data processing techniques that account for missing values.

A further limitation concerns the fact that the autistic participants were highly verbal, and as such, may not be representative of the spectrum in its full range. Thus, the current results may reflect performance primarily by highly verbal children and young adults with autism. Future research should focus on the full range of verbal ability whereby the eye-tracking approach may be even more telling for the processing of social cues.

## Figures and Tables

**Figure 1 behavsci-15-01622-f001:**
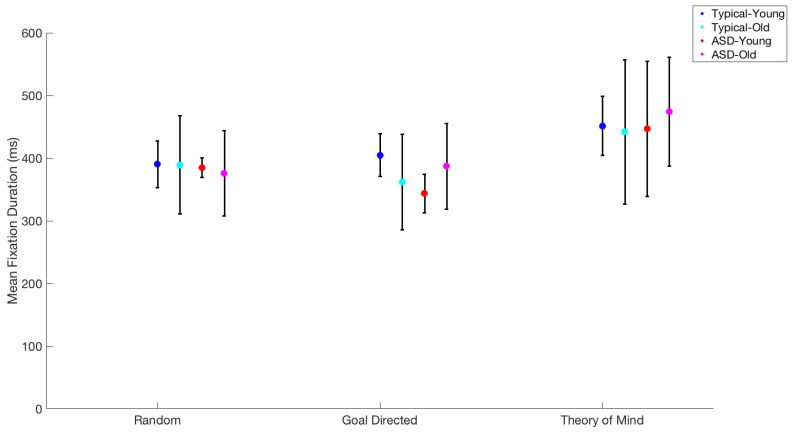
Mean fixation duration for each age group and type of expression following the work of [19] ([19]) and [39] ([39], [40]). Error bars indicate standard error of the mean.

**Figure 2 behavsci-15-01622-f002:**
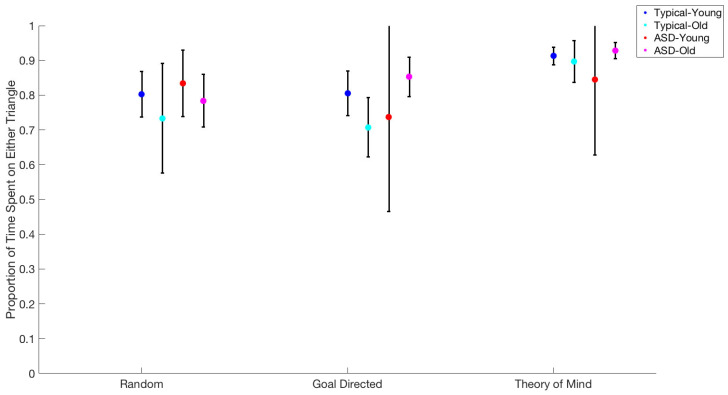
Proportion of time spent on either triangle for each age group for each movie type.

**Figure 3 behavsci-15-01622-f003:**
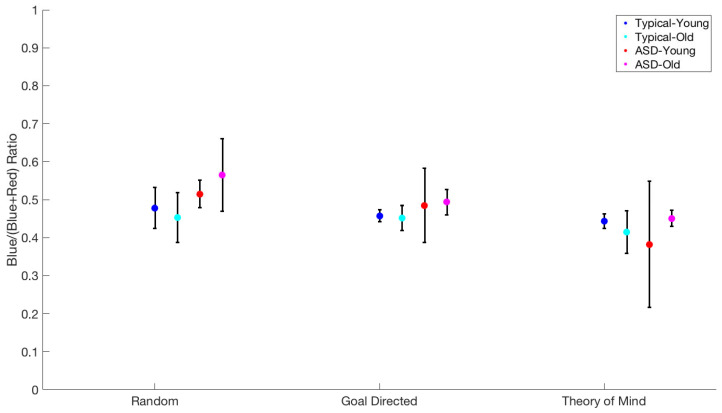
Blue vs. red ratio for each age group for every type of movie.

**Figure 4 behavsci-15-01622-f004:**
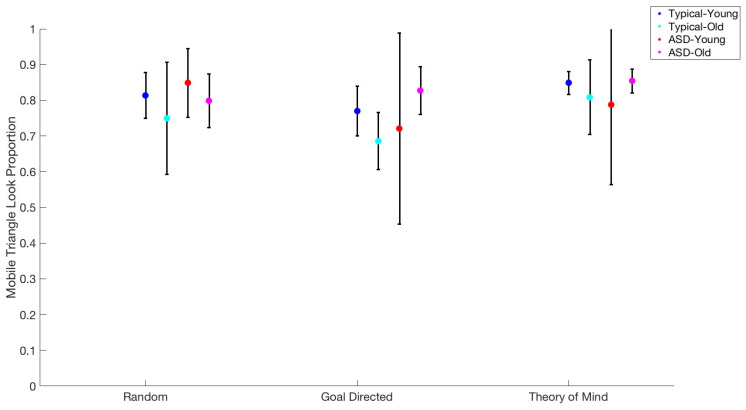
Mobile triangles look proportion for each age group for every type of movie.

**Figure 5 behavsci-15-01622-f005:**
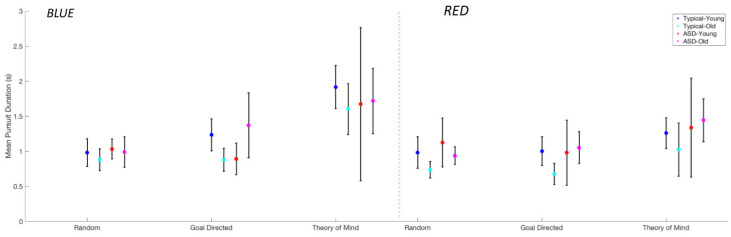
Mean pursuit duration per triangle for each group and each type of movie.

**Table 1 behavsci-15-01622-t001:** Mean verbal responses per age and group (ASD vs. TD).

Children (10–13 Years Old)				Young Adults (14–23 Years Old)			
	ToM	Goal-Directed	Random		ToM	Goal-Directed	Random
Intentionality							
ASD	2.90 *^,§^	2.31	0.65	ASD	3.57 ^§^	2.24	0.35
TD	3.36 *	2.30	1.16	TD	3.56	2.33	0.80
Appropriateness							
ASD	0.83 ^§^	1.47	1.09	ASD	1.45	1.37	1.21 ^§^
TD	1.14	1.58	1.46	TD	1.71	1.51	1.36
Length							
ASD	2.68 *^,§^	2.33 ^§^	2.17 ^§^	ASD	3.29 ^§^	2.86 ^§^	2.70 ^§^
TD	3.26 *	2.62	2.33	TD	3.42	2.65	2.10

Note. * *p* < 0.05 within Age; ^§^ *p* < 0.05 within Group. The score ranges were 0 to 5 for intentionality, 0 to 3 for appropriateness, and 0 to 4 for the length of verbal responses.

**Table 2 behavsci-15-01622-t002:** GLM Results for mean smooth pursuit durations.

Effect	F	Hypothesis df	Error df	Sig.	Partial Eta Squared	Observed Power
MovieType	21.35	2.00	15.00	<0.001	0.74	1.00
MovieType × asd	0.33	2.00	15.00	0.73	0.04	0.09
MovieType × age_binary	0.18	2.00	15.00	0.84	0.02	0.07
MovieType × asd × age_binary	0.80	2.00	15.00	0.47	0.10	0.16
TriangleType	16.00	1.00	16.00	0.001	0.50	0.96
TriangleType × asd	2.54	1.00	16.00	0.13	0.14	0.32
TriangleType × age_binary	0.63	1.00	16.00	0.44	0.04	0.12
TriangleType × asd × age_binary	0.58	1.00	16.00	0.46	0.03	0.11
MovieType × TriangleType	6.26	2.00	15.00	0.01	0.45	0.82
MovieType × TriangleType × asd	0.13	2.00	15.00	0.88	0.02	0.07
MovieType × TriangleType × age_binary	1.04	2.00	15.00	0.38	0.12	0.20
MovieType × TriangleType × asd × age_binary	1.11	2.00	15.00	0.36	0.13	0.21

## Data Availability

The data presented in this study are available on request from the corresponding author.

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
