# Peer review of "Windows to the Social Mind: What Eye-Tracking Reveals About Theory of Mind in Children and Young Adults with Autism Spectrum Disorder (ASD)"

_behavsci, 2025, doi:10.3390/bs15121622_

Round 1

Reviewer 1 Report

Comments and Suggestions for Authors

The manuscript is well written, theoretically grounded, and clearly structured. The introduction provides sufficient context on Theory of Mind (ToM) and Autism Spectrum Disorder (ASD), and the methods and results are transparently reported. The findings make a modest but clear contribution by showing that while eye-movement patterns appear typical in ASD, verbal ToM descriptions differ by age and diagnosis. The language is polished and professional.

However, several areas could be strengthened to enhance rigor and reader clarity.

  1. Literature and Framing
    The authors integrate key foundational studies (Frith & Happé, Baron-Cohen, etc.).The introduction is very detailed but perhaps too dependent on older literature (mostly pre-2015). Adding 2–3 recent ToM or eye-tracking studies (2022–2024) could help show the paper’s relevance to current debates, especially around ecological validity and multimodal ToM measurement.
  2. Citation (“Jones et al., 2026”)
    The reference “Jones et al., 2026” appears to be an error. The likely The 2026 reference should be corrected.
  3. The paper would benefit from more detail on eye-tracking preprocessing (e.g., filtering parameters, exclusion criteria). While some details are provided (dispersion algorithm, thresholds), it’s not fully clear how artifacts and missing samples were handled.
  4. Practical Implications
    The discussion is well reasoned but could briefly expand on clinical or educational implications, e.g., how eye-tracking might support assessment of ToM in minimally verbal children.

Author Response

Reviewer 1

The manuscript is well written, theoretically grounded, and clearly structured. The introduction provides sufficient context on Theory of Mind (ToM) and Autism Spectrum Disorder (ASD), and the methods and results are transparently reported. The findings make a modest but clear contribution by showing that while eye-movement patterns appear typical in ASD, verbal ToM descriptions differ by age and diagnosis. The language is polished and professional.

Response:
We thank the reviewer for this encouraging feedback. We have revised the manuscript in accordance with all suggestions.

However, several areas could be strengthened to enhance rigor and reader clarity.

  1. Literature and Framing

The authors integrate key foundational studies (Frith & Happé, Baron-Cohen, etc.). The introduction is very detailed but perhaps too dependent on older literature (mostly pre-2015). Adding 2–3 recent ToM or eye-tracking studies (2022–2024) could help show the paper’s relevance to current debates, especially around ecological validity and multimodal ToM measurement.

Response:
We thank the reviewer for this helpful suggestion. In the revised manuscript, we have added recent studies to strengthen the theoretical framing and highlight the paper’s relevance to current debates on ecological validity and multimodal ToM measurement. Specifically, we now cite Wohltjen and Wheatley (2024), who used interpersonal eye-tracking in live social interactions to examine dynamic gaze coupling, and Ford et al. (2024), who employed eye-tracking to explore visual perspective-taking and attentional modulation by social context. These additions situate our study within the most recent developments in Theory of Mind and eye-tracking research.

  1. Citation (“Jones et al., 2026”)

The reference “Jones et al., 2026” appears to be an error.

Response:
We thank the reviewer for catching this error. It has been corrected to Jones et al., 2018.

  1. Eye-tracking preprocessing details

The paper would benefit from more detail on eye-tracking preprocessing (e.g., filtering parameters, exclusion criteria). While some details are provided (dispersion algorithm, thresholds), it is not fully clear how artifacts and missing samples were handled.

Response:

We appreciate the reviewer’s observation regarding eye-tracking preprocessing. We have expanded the Methods section (Section 3.2) to clarify the preprocessing pipeline, including filtering parameters, handling of artifacts and missing samples, and specific exclusion criteria. The revised text now details our interpolation procedure, median-filter settings, and quality-control thresholds to ensure data reliability.

  1. Practical Implications

The discussion is well reasoned but could briefly expand on clinical or educational implications, e.g., how eye-tracking might support assessment of ToM in minimally verbal children.

Response:
We thank the reviewer for this insightful suggestion. We have added the following text:

“In particular, eye-tracking can offer valuable insights into how children attend to and process social cues, even in the absence of verbal responses. This makes it a promising tool for assessing ToM in minimally verbal children, where traditional language-based assessments may be inadequate or biased. From a clinical perspective, eye-tracking can help identify atypical social attention patterns and guide individualized interventions. In educational settings, it may inform tailored support by revealing each child’s social-cognitive strengths and needs.” 

Reviewer 2 Report

Comments and Suggestions for Authors

This is an interesting paper that will make a substantial contribution to the literature. By bringing together both verbal data and eye tracking data, you are able to provide greater insight into the field of ToM and specifically the use of eye tracking as a means of understanding ToM in people living with autism. On the whole I think this manuscript publishable in its current form. There are a few very minor additions that I would have liked to have seen, but these are personal preferences.

1. You provide a comprehensive review of literature connected to ToM, but only briefly touch on autism itself. It would have been nice to see some background information specific to the broader field before you launch into the focus of your paper.

2. I also found that a little more description in places would have been useful. For example, when you note the complication with the 'false belief tasks' it would have been good for you to be specific about what this task involves and the specific features that make it susceptible to confounding factors.

3.I also note some typesetting issues throughout, and these would need to be addressed prior to publication.

Author Response

Reviewer 2

This is an interesting paper that makes a substantial contribution to the literature. By combining verbal data and eye-tracking data, the study provides greater insight into ToM and highlights the value of eye-tracking for understanding ToM in autism. On the whole, the manuscript is publishable in its current form. There are only a few very minor additions that would enhance it.

Response:
We sincerely thank the reviewer for their encouraging feedback. We have revised the manuscript according to all suggestions.

  1. Autism background

You provide a comprehensive review of literature connected to ToM but only briefly touch on autism itself.

Response:
We thank the reviewer for this constructive comment. The Introduction has been expanded to include a concise overview of Autism Spectrum Disorder (ASD) and its relevance to social cognition research. This new paragraph outlines the core features of ASD, the variability in cognitive and linguistic profiles, and how these characteristics relate to ToM development and assessment.

  1. False-belief task description

The paper should specify what the false-belief task involves and the factors that make it susceptible to confounding effects.

Response:
We appreciate this suggestion and have expanded the Introduction to briefly describe the false-belief paradigm and its limitations, including the influence of language demands, executive function, and inhibitory control. This provides clearer context for the use of eye-tracking as an alternative, more objective method for assessing ToM.

  1. Typesetting issues

Some typesetting inconsistencies were noted throughout.

Response:
We thank the reviewer for noting these issues. The manuscript has been carefully proofread and reformatted to correct spacing, alignment, and font inconsistencies. Heading styles, table and figure captions, and reference formatting have been standardized in line with journal guidelines.

Reviewer 3 Report

Comments and Suggestions for Authors

Dear Authors,

Thank you for submitting your manuscript titled “Windows to the Social Mind: What Eye-Tracking Reveals About Theory of Mind in Children and Young Adults with Autism Spectrum Disorder” for consideration in the journal Behavioural Sciences. I appreciate the opportunity to review your work, which makes an important contribution.

Your paper is well written, theoretically grounded, and methodologically sound. However, I would like to offer several suggestions that may help strengthen the manuscript and increase its clarity and impact.

Title

The title is clear, informative.

Abstract

The abstract is concise but could be improved by stating the sample size and key statistical results to enhance transparency.

Use consistently the abbreviations e.g., “ASD,” “TD

Introduction

The introduction mentions that eye-tracking is “more objective,” but it should specify why it is theoretically meaningful in ToM research.

While Heider & Simmel (1944) is foundational, one brief reference is sufficient; the full description of the experiment can be shortened.

Materials and method

Specify whether participants had comorbid conditions (e.g., ADHD, language delay) and if IQ or cognitive ability was controlled for, as these can strongly influence ToM and verbal performance.

Indicate which eye-tracking variables were analyzed (e.g., fixation duration, saccade length, dwell time on agents, gaze transitions). Currently, this is not specified. Include total animation duration, frame rate, and whether the order of animations was counterbalanced between participants.

The procedure section is well written.

Results

The structure of results section is logical, progressing from behavioral to physiological data and ending with integrative models.

Discussion

The Discussion section provides a well-organized summary of the study’s main findings, however, it remains somewhat descriptive rather than interpretative. Discuss the results in the context of major theoretical frameworks such as dual-process models of ToM.

Conclusion

The conclusion is clear, relevant, and aligned with the study’s aims.

References

Well-referenced.

Author Response

Reviewer 3

Dear Authors,

Thank you for submitting your manuscript “Windows to the Social Mind: What Eye-Tracking Reveals About Theory of Mind in Children and Young Adults with Autism Spectrum Disorder” for consideration in Behavioral Sciences. Your paper is well written, theoretically grounded, and methodologically sound. I offer several suggestions to further strengthen its clarity and impact.

Response:
We thank the reviewer for these valuable suggestions and have revised the manuscript accordingly.

Title
Clear and informative.

Abstract
The abstract should include sample size and key statistical results to enhance transparency.

Response:
We thank the reviewer and have added the sample size and key statistical indicators to the abstract.

Abbreviations (ASD, TD)

Ensure consistent use of abbreviations throughout.

Response:
We appreciate this observation. The manuscript has been carefully reviewed to ensure consistent use of abbreviations such as Autism Spectrum Disorder (ASD) and Typically Developing (TD). Full terms are spelled out at first mention in each section, followed by the abbreviation in parentheses, per journal style.

Introduction
Clarify why eye-tracking is theoretically meaningful in ToM research.

Response:
We have revised the text as follows:

“White, Coniston, Rogers, and Frith (2011) developed a more objective approach to capture spontaneous, nonverbal indicators of social attention and mental-state attribution by tracking gaze. By revealing how individuals allocate visual attention to intentional versus random motion, eye-tracking allows researchers to infer implicit ToM processing beyond verbal or explicit responses.”

Also, the description of Heider & Simmel (1944) can be shortened.

Response:
We have shortened the description accordingly.

Materials and Methods

Specify comorbidities and cognitive controls.

Response:
We have clarified in the Participants subsection that none of the participants had comorbid conditions (e.g., ADHD, language delay) and that all had age-appropriate cognitive and verbal abilities. Standardized cognitive assessments were used where applicable.

Specify which eye-tracking variables were analyzed, and include animation duration, frame rate, and counterbalancing.

Response:
The Eye-Tracking Analysis subsection has been expanded to include details on fixation duration, saccade length, dwell time on agents and targets, and gaze transitions. Total animation duration (40 s), frame rate (30 fps), and counterbalancing procedures have been specified.

Results
Clear and logically structured.

Discussion
The Discussion is well organized but somewhat descriptive. Please discuss results within major theoretical frameworks such as dual-process models of ToM.

Response:
We thank the reviewer for this valuable comment. The Discussion has been revised to include a stronger interpretative section linking findings to implicit versus explicit aspects of ToM. The revised paragraph highlights that individuals with ASD may show preserved implicit social attention (eye-tracking data) but reduced explicit reasoning (verbal ToM performance), consistent with distinctions described in earlier theoretical accounts (Frith & Happé, 1994; Baron-Cohen, 1995; Chahboun, 2017; Erena-Guardia et al., 2024).

Conclusion
Clear, relevant, and aligned with the study’s aims.

References
Comprehensive and appropriate.